# Association between Smoking and Urine Indole Levels Measured by a Commercialized Test

**DOI:** 10.3390/metabo12030234

**Published:** 2022-03-09

**Authors:** Masataka Mine, Nobuyuki Masaki, Takumi Toya, Takayuki Namba, Yuji Nagatomo, Bonpei Takase, Takeshi Adachi

**Affiliations:** 1Japan Air Self-Defense Force Aeromedical Laboratory, Sayama 350-1324, Japan; arzt.mine@gmail.com; 2Department of Intensive Care Medicine, National Defense Medical College, Tokorozawa 359-8513, Japan; con367@ndmc.ac.jp (T.T.); dui15772004@yahoo.co.jp (B.T.); 3Department of Cardiology, National Defense Medical College, Tokorozawa 359-8513, Japan; con079@ndmc.ac.jp (T.N.); con401@ndmc.ac.jp (Y.N.); kenada@ndmc.ac.jp (T.A.)

**Keywords:** metabolism, indole, smoking, clinical studies

## Abstract

Indoles are formed from dietary tryptophan by tryptophanase-positive bacterium. A few amounts of indole are excreted in the urine. On the other hand, cigarette smoke contains indoles, which could also change the urine indole levels. This study sought to elucidate the relationship between urine indole levels and smoking habits. A total of 273 healthy men (46 ± 6 years old) were enrolled in the study. Fasting urine and blood samples were obtained in the morning. The indole concentration was measured by a commercialized kit with a modified Kovac’s reagent. The relationship with smoking status was evaluated. The median value of the urine indole test was 29.2 mg/L (interquartile range; 19.6–40.8). The urine indole level was significantly elevated in the smoking subjects (non-smoking group, 28.9 (20.9–39.1) mg/L, *n* = 94; past-smoking group, 24.5 (15.7–35.5) mg/L, *n* = 108; current-smoking group, 34.3 (26.9–45.0) mg/L, *n* = 71). In the current-smoking group, urine indole levels correlated with the number of cigarettes per day (ρ = 0.224, *p* = 0.060). A multivariate regression test with stepwise method revealed that the factors relating to urine indole level were current smoking (yes 1/no 0) (standardized coefficient β = 0.173, *p* = 0.004), blood urea nitrogen (β = 0.152, *p* = 0.011), and triglyceride (β = −0.116, *p* = 0.051). The result suggests that smoking is associated with increased urine indole levels. The practical test might be used as a screening tool to identify the harmful effect of smoking.

## 1. Introduction

Indoles are substances metabolized from dietary tryptophan by gut bacteria [1]. The production requires the action of bacterial tryptophanase, which is expressed in *Escherichia coli*, *Clostridium* spp., and *Bacteroides* spp. [2,3]. Indoles play diverse biological roles for some bacteria, including spore formation, drug resistance, virulence, plasmid stability, and biofilm formation [2]. On the other hand, indoles within physiological fecal concentration have beneficial effects on the epithelium of the intestine, such as anti-inflammatory effects and acting as a mucosal barrier [4]. Indoles are possibly absorbed into body from outside. Indoles are contained in mainstream cigarettes [5,6].

Indoles are solid at body temperature, but soluble in water at low concentration. Indoles are absorbed efficiently through the epithelium of the jejunum, ileum, and colon, and enter circulation [7]. Indoles are further oxidized and conjugated with sulfate to become indoxyl sulfate in the liver and eliminated into the urine [8]. Several studies have confirmed that indoles are also excreted into urine in their original form or glucuronidation conjugates by mass spectrum [9,10]. A previous study using [2-^14^C] indole in rats demonstrated that orally administrated indole was rapidly excreted, and in 2 days an average of 81% appeared in the urine, 11% in the feces, and 2–4% in the expired air as carbon dioxide [11]. The radioactivity is excreted in the urine as 50% of the indoxyl sulfate and 0.07% as unchanged indole. The great majority of fecal indole concentrations have been found to fall within the range from 1.0 to 4.0 mM in humans [12]. However, normal concentrations of indoles in human urine has not been sufficiently acknowledged [9].

Recently, a convenient test has been developed that measures urine indole concentrations by a colorimetric technique with a modified Kovac’s reagent. Indole test has been originally used to distinguish *Escherichia coli* in bacterial culture examinations [12,13,14]. The urine indole test has yet to be evaluated in clinical studies to the extent it deserves. Although a liquid/mass spectrometer can provide precise information on metabolites in urine, the method needs time and effort. The convenient test might be useful as a screening tool. Therefore, we sought to investigate the association between urine indole levels and lifestyle, especially smoking habits, using the measurement kit. The results might help to identify the effect of smoking on health.

## 2. Results

### 2.1. Clinical Characteristics

Table 1 shows the characteristics of the participants (*n* = 273) divided into the quartile groups by urine indole concentration levels. The average age was 46 ± 6 years old. The ratio for current smoker increased with the quartile rank (Figure 1). The highest quartile group had higher blood urea nitrogen (BUN), and lower alanine aminotransferase, γ-glutamyltransferase, and triglyceride levels. No participant was urine protein positive or urine glucose positive. The subjects were also divided into the three groups by smoking status (Appendix A). The past-smoking group was older than those in the non-smoking group who had never smoked before. Alcohol intake was higher in the current-smoking group than the non-smoking group.

### 2.2. Urine Indole Levels and Smoking Status

The median urine indole level for all participants was 29.2 (19.6–40.8) mg/L. The urine indole levels for the current-smoking group were higher than those for the past-smoking group and non-smoking group (non-smoking group 28.9 (20.9–39.1) mg/L, past-smoking group 24.5 (15.7–35.5), current-smoking group 34.3 (26.9–45.0) mg/L, *p* = 0.001). (Figure 2).

In the post-hoc analysis, there was a significant difference in urine indole levels between the current-smoking group and non-smoking group (*p* = 0.013), or the past-smoking group (*p* < 0.001). In the current-smoking group, urine indole levels correlated with the number of cigarettes per day (ρ = 0.224, *n* = 71, *p* = 0.060).

We further investigated whether the types of cigarettes affected the urine indole levels in the current-smoking group (Figure 3). The urine indole concentrations did not differ between the smokers using traditional paper cigarettes and electronic cigarettes (e-cigarette) (paper cigarettes, 33.2 (25.4–44.2) mg/L, *n* = 55; e-cigarettes, 42.3 (28.4–52.4) mg/L, *n* = 16, *p* = 0.271).

### 2.3. Univariate and Multivariate Regression Analyses for Urine Indole Concentration

Univariate regression analysis showed the urine indole concentration level negatively associated with serum triglycerides and alanine aminotransferase, although body mass index was not associated (Table 2). In addition, the urine indole levels were positively associated with BUN and creatinine kinase serum levels. We also quantified alcohol consumption, but any correlation with urine indole levels was not detected. The results of univariate analysis with other factors are additionally shown in Appendix A. In multivariate analysis, current smoking is related to urine indole levels after adjustment for the co-factors in both the crude model (*p* = 0.011) and the stepwise model (*p* = 0.004).

### 2.4. Diurnal Variations and Effect of Smoking

Twelve subjects comprising four smokers of paper cigarettes (A1–4), four smokers of electric smokers (B1–4), and four non-smokers (C1–4) were enrolled (Table 3). The average CV of the three time points was 15.4 ± 12.2%. The intra-rater reliability for the three repetitive measurements was 0.781, *p* = 0.001. There was no statistical difference in the concentration of urine indole among the three time points (*p* = 0.344). In the eight smokers, the smoking effect on the urine tests (between Time 2 and Time 3) was not observed (*p* = 0.111).

## 3. Discussion

This study revealed that urine indole levels in the current-smoking group were higher than those in the non-smoking group and past-smoking group. The indole concentration of past-smokers was as low as that of controls who had never smoked, suggesting that the influence from smoking ends with quitting. Some participants without smoking habits had high urinary indole concentrations whereas the ratio of past-smokers decreased with the rank of quartiles. The reason was unknown but influencing factors other than smoking has been suggested. Additionally, urine indole levels were not different between the current-smokers using e-cigarettes and paper cigarettes. Because there have been few studies investigating indole levels in urine [9,10], this is the first report to our knowledge suggesting increased urinary indole levels measured by a commercialized kit using the Kovac’s reagent and smoking status.

Several mechanisms to increase urine indole levels are known. The possible endogenous reasons include hepatic dysfunction and increases in bacterial indole production. Serum indole levels rise together with ammonia in patients with hepatic encephalopathy because the detoxication of indole to indoxyl sulfate in the liver is impaired [15]. However, because the study enrolled only healthy participants, this reason is unlikely. The diagnosis of NAFLD was not associated with urine indole concentration levels.

On the other hand, indole is a molecule not produced in human metabolism but is produced in tryptophanase-positive bacteria living in the intestine [16]. It is therefore speculated that smoking could make a gut environment favorable to such bacteria for growth and activity. Up to 85 indole-producing bacteria have been previously identified [2], but most of them are minor gut colonists or low-abundance pathogens [17]. *Escherichia coli* is the most famous indole-producing bacterium. However, research on putative tryptophanases in the genome sequences has revealed that tryptophanases in *Bacteroides* were much more abundant than those from other origins [1]. For example, *Bacteroides thetaiotaomicron* is one of the main producers of indole [2,16] and short-chain fatty acids, which have anti-inflammatory properties in the intestinal mucosa [18]. Thus, *Bacteroides* are reported to be major bacteria for indole production in the intestine.

Few studies investigating the effects of cigarette-smoke on intestinal microflora have been previously conducted [19]. Smokers are known to be susceptible to *Clostridium difficile* infection [20]. In a recent cross-sectional study, the intestinal microflora in current smokers was characterized by higher *Bacteroidetes*, and lower *Firmicutes* and *Proteobacteria* in the philias level [21]. Interestingly, the mean ratio of *Firmicutes*/*Bacteroidetes* in former smokers was not different from that of the controls who had never smoked. This is the same as the urine indole concentrations in our study. Moreover, longitudinal studies have demonstrated that smoking cessation increases microbial diversity and proportions in *Firmicutes* to *Bacteroidetes* [22,23]. In general, smoking induces diet changes and weight loss [24]. *Firmicutes* is supposed to be a pathogenesis of weight gain after smoking cessation. Carbohydrates are preferentially fermented by gut bacteria that may result in the decreased production in indoles from tryptophan degradation [25]. Conversely, a high-protein diet instead of carbohydrates upregulates tryptophanase activity in *Escherichia coli* [26]. Yogurt whey decreases indole production by *Escherichia coli* [27]. Urinary levels of indoles may reflect the changes in intestinal bacteria. For instance, urinary 3-indoxyl sulfate was already reported as a marker for gut microbiota diversity [28]. It decreased when gut microbiota was disrupted by extensive use of antibiotics after allogeneic stem cell transplantation [29].

Another possibility is the detection of compounds, such as indole, in tobacco. The specificity of the modified Kovac’s reagent to indole is also to be considered. Riveles et al. have reported the presence of indole and 5-methylindole in mainstream tobacco smoke [5], but their absorption from the lungs into the circulation is unknown. In our longitudinal evaluation, indole concentrations measured with the commercial kit did not decrease with sleep or increase after smoking (Table 3), suggesting that the direct effect of smoking is not significant. The indole test may also respond to other substances similar to indoles. 3-methylindole (skatole) is another bacterial metabolite from tryptophan [30,31]. Skatole is also found in tobacco and is a potential lung carcinogen [32]. A conventional Kovac’s reagent [4-(Dimethylamino)benzaldehyde] reacts with skatole, but the reaction is much smaller than that with indole. [12] In addition, the Kovac’s reagent reacts less well with indole-3-propionic acid, indole-3-pyruvic acid, and indoxyl sulfate [12], which are naturally occurring indole analogues present in biological samples [4]. Urinary 2-amino-9H-pyrido[2,3-b]indole (AαC), a heterocyclic aromatic amine (HCA), is known to be increased in smokers [33]. Cigarettes contain other HCAs, such as 3-amino-1,4-dimethyl-5H-pyrido[4,3-b]indole (Trp-P-1) and 3-amino-1-methyl-5H-pyrido[4,3-b]indole (Trp-P-2) [34,35]. These HCAs are not found in unburned tobacco [36] and are produced by heating tobacco filters [35,36]. Normal levels of HCAs in tobacco smoke have been reported to be produced at 0.3–260.0 ng per cigarette [37], so the final levels excreted in urine are very low. It is not known whether the modified Kovac’s reagent reacts to HCAs.

Thus, our results suggest the possibility that toxic chemical compounds in cigarettes and a change of diet to fewer carbohydrates increase composition or activity of tryptophanase-positive bacteria leading to increased indole production. The urinary indole levels may also be influenced by absorption of indoles from smoking. A direct comparison of urine indole levels with fecal amounts of indoles and tryptophanase-positive bacteria is necessary to prove the involvement of intestinal microflora in the future. The study poses a question on whether the condition rising urine indole levels is associated with the smoking-caused diseases. It has been reported that indoles have a beneficial effect on intestinal health for anti-inflammatory properties [4]. Nonetheless, indole and 5-methylindole are the most potent and efficacious cigarette smoke components tested in an animal study, which adversely affect several reproductive processes, including oviductal functioning [5]. Although e-cigarettes have become more popular these days, cardiovascular risk [38,39] and influence on gut microbiota [40] have been reported. Urine indole levels were not different between traditional paper cigarettes and e-cigarettes in the study. This may suggest the e-cigarettes do not reduce the effects of smoking on intestinal microflora.

There are several limitations to this study. First, smoking status was confirmed by self-assessment with written forms. There was the possibility that some participants might not have revealed their smoking behavior. It is ideal to measure smoking status objectively, such as urine nicotine/cotinine level; however, the assessment could not be performed in the study. Second, the urine test may be affected by urinary tract infection; however, there were no participants with ongoing treatment of urinary tract infections. Third, urine creatinine was not measured together with indole in this study. It has been recommended to calculate analyte/creatinine ratios at quantitative determinations in spot urine to correct diuresis [41]. However, the urine samples were obtained in the same regulated conditions. As shown in Table 3, the indole test values were most likely due to individual differences, with the exception of diuresis. Fourth, the study enrolled only healthy middle-aged men. It is uncertain if the results would be the same in other populations. However, the restricted study population is strength for elucidating the effect of smoking by excluding other clinical factors.

In conclusion, current smokers had higher urine indole levels. This may reflect changes in the intestinal microflora by smoking. It is certain that liquid/mass spectrum is required to measure accurate concentrations of urine metabolites; the commercialized test utilizing the modified Kovac’s reagent could be used as screening for the purpose of detecting the harmful effects of smoking.

## 4. Material and Methods

### 4.1. Study Design and Population

We enrolled middle-age healthy men (40–55 years old) who received annual check-ups and agreed to participate in a survey on metabolic syndrome at Iruma Air Defense Force Base (Saitama, Japan). This clinical study conformed to the ethical guidelines of the Declaration of Helsinki and was approved by the ethics committee at our institute (Registration No. 1–3). Written informed consent was obtained from all participants. The exclusion criteria were diabetes mellitus, uncontrolled hypertension and/or dyslipidemia, chronic kidney disease, hemodialysis, infections, medication of antibiotics, *Lactobacillus* preparation, malignant tumor, and cardiovascular disease. Cross-sectional analyses were performed on an annual dataset. We added the study for ensuring diurnal variations and the effect of smoking on the urine indole test.

### 4.2. Anthropometric and Biochemical Data

Data were collected at a single facility. Sociodemographic information, such as age, were collected from the database before the examinations. Weight and height were measured using standard methods and digitally recorded. Chest circumferences were measured just below the nipples. Waist circumferences were measured at the navel as participants exhaled lightly while standing. Blood pressure was measured around the upper arm with participants in a sitting position after a few minutes of rest. Life history, including alcohol intake, smoking, and medications, was investigated by a questionnaire.

Fasting blood and urine samples were obtained in the morning on the day when a gastric fiberscope was scheduled for annual health check. All participants were asked to fast from the previous evening. Blood and urine specimens were immediately processed at the same facility. The urine indole test (Wismerll Co., Ltd., Tokyo, Japan) (Diacron, Grosseto, Italy) measures colorimetric changes in light absorbance at 540 nm wavelength 5 min after adding 50 µL of the reagent into 1 mL of the urine sample. The main substances in the modified Kovac’s reagent are 1-pentanol (Conc.% *w*/*w*, <30%) and hydrochloric acid (Conc.% *w*/*w*, <90%). The temperature of the reagent and sample was maintained at 37 °C before testing. The normal concentration was reported to be under 20 mg/L (171 µM). The coefficient of variation (CV) was reported to be 4.4 ± 2.7% by the manufacture.

Non-alcoholic fatty liver disease (NAFLD) was diagnosed by abdominal ultrasound as follows. After skilled laboratory technicians performed a primary assessment, a gastroenterologist conducted a second evaluation and reached a diagnosis. Metabolic syndrome was defined as the presence of three or more of the following five abnormalities: (1) abnormal waist circumference (male ≥ 85 cm, female ≥ 90 cm), (2) fasting serum triglycerides ≥ 150 mg/dL, (3) HDL < 40 mg/dL, (4) fasting blood glucose ≥ 110 mg/dL, and (5) blood pressure ≥ 130/85 mmHg [42]. The homeostasis model assessment for insulin resistance index and insulin secretion ability was assessed based on fasting insulin levels as previously described [43].

### 4.3. Diurnal Variations and Effect of Smoking

The diurnal variations and the effect of smoking on indole urine concentrations was examined separately from the first cross-sectional study. The middle-age healthy men were recruited for the study. All participants were asked to fast from the previous evening. Urine samples were collected at three time points: the night before sleep (day 1), in the morning, and a few hours later (day 2). The latter two were before and after smoking in the case of smokers. The first timing on day 2 was the same as when the cross-sectional study was conducted. Urine specimens were kept in cold storage until measurement, and all urine concentrations were measured together on the same day.

### 4.4. Statistics

The distribution of measurements was evaluated by examining a histogram and applying the Shapiro–Wilk test. The independent sample *t* test or Mann–Whitney *U* test was used for two-group comparisons. For comparison among more than three groups, analysis of variance (ANOVA) with a post-hoc test using the Bonferroni method and Kruskal–Wallis test were carried out for parametrical and non-parametrical variables, respectively. Categorical clinical characteristics were compared using Chi-square testing or Fisher’s exact test, if appropriate. The correlation efficient among two variables was obtained with Pearson’s method or Spearman’s method, if appropriate.

Univariate and multivariate regression analyses were performed to identify clinical factors associated with urine indole levels. In the multivariate analysis, age, body mass index, alcohol consumption, hypertension, hyperlipidemia, NAFLD, and the independent factors correlating with urine indole levels (<0.1) in the univariate analysis were included in a crude model (model 1). Next, a stepwise method was used to select effective explanatory variables from the variables used in model 1. Unnecessary variables were removed to obtain a well-fitting model (model 2).

In the additional study for repetitive measurements, inter-class correlation was calculated for evaluation of the intra-rater reliability of the indole test. The difference in measurement values between the times was also analyzed by repeated measure analysis of variance. The difference before and after smoking in smokers was analyzed by paired t test. Statistical analyses were performed using SPSS version 22.0 (SPSS, Japan, Tokyo). Summary data are presented as means ± standard deviations or median (1st and 3rd quartiles). In all analyses, *p* < 0.05 was considered statistically significant.

## Figures and Tables

**Figure 1 metabolites-12-00234-f001:**
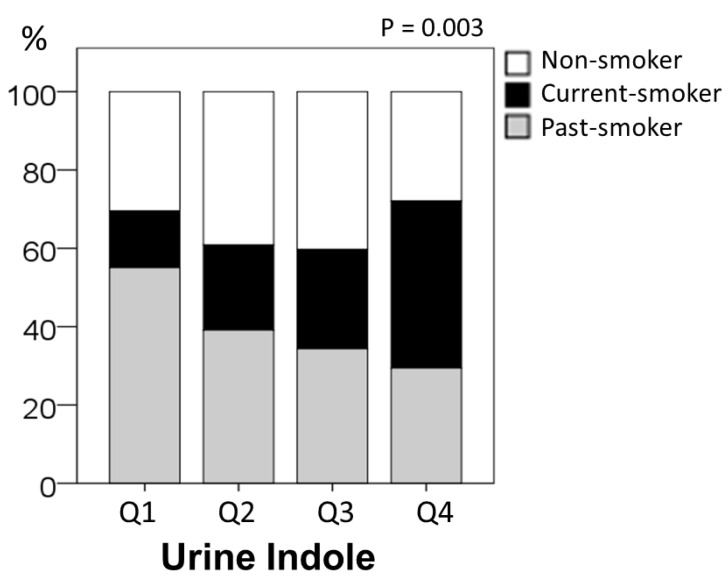
The ratio of smoking status in the quartile groups by urine indole concentration level. The *p*-value on the graph stands for the statistical significance of difference in the proportion of the three smoking groups among the quartile groups (*p* = 0.003).

**Figure 2 metabolites-12-00234-f002:**
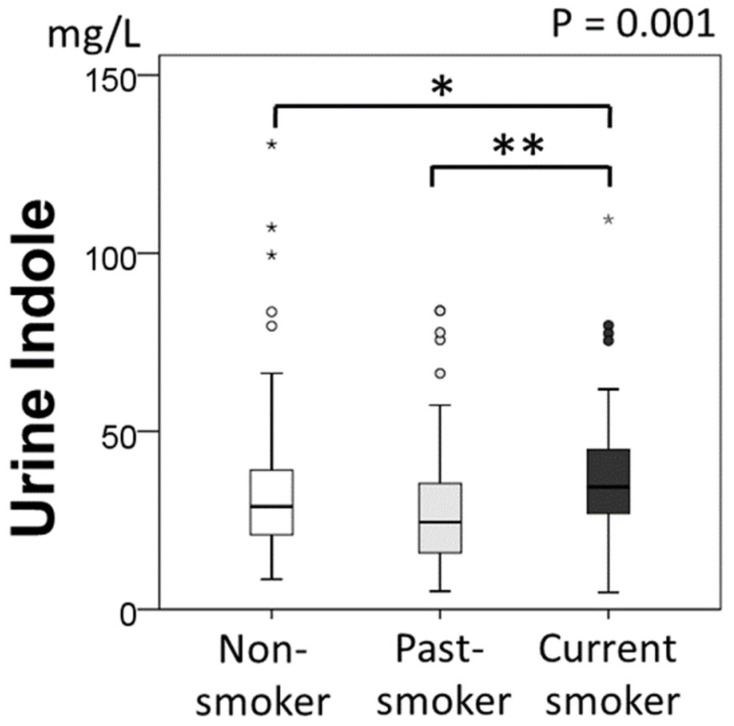
Comparison of urine indole levels by smoking status. The *p*-value on the graph stands for the statistical significance of difference among three groups (*p* = 0.001). * *p* < 0.05, ** *p* < 0.01.

**Figure 3 metabolites-12-00234-f003:**
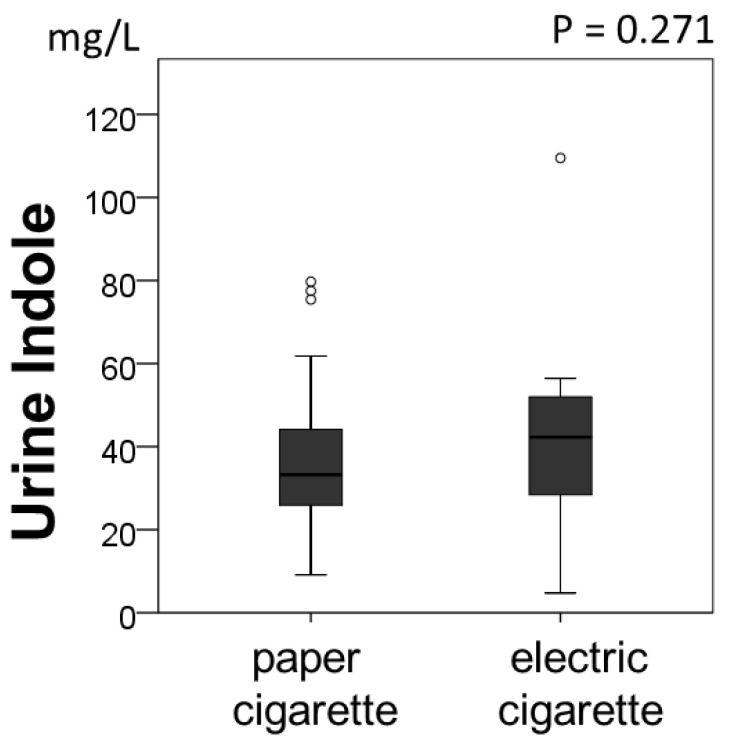
Comparison of the types of cigarettes in urine indole levels for current-smokers. The *p*-value on the graph indicates that there is no significant difference among the groups (*p* = 0.271).

**Table 1 metabolites-12-00234-t001:** Background characteristics of the subjects.

			Urine Indole Concentration (mg/L)			
		Q1	Q2	Q3	Q4	
		≤19.73	19.74–29.19	29.20–40.78	40.79+	
**(No. of Subjects)**	(273)	(69)	(69)	(67)	(68)	*p*-Value
Age (years)	45.7 ± 5.9	46.1 ± 4.7	46.6 ± 5.6	45.1 ± 6.6	45.1 ± 6.7	0.365
BMI (kg/m^2^)	23.7 ± 2.2	23.9 ± 2.0	23.4 ± 2.0	23.7 ± 2.3	23.7 ± 2.3	0.592
Circumference of Chest (cm)	94.8 ± 5.3	94.7 ± 5.1	94.3 ± 5.1	95.3 ± 5.5	94.8 ± 5.7	0.752
Circumference of Abdomen (cm)	82.7 ± 6.1	83.6 ± 5.2	82.1 ± 5.3	82.5 ± 7.2	82.7 ± 6.3	0.531
Systolic Blood Pressure (mmHg)	116.3 ± 11.6	114.9 ± 10.9	115.7 ± 11.6	116.7 ± 12.5	117.8 ± 11.5	0.499
Diastolic Blood Pressure (mmHg)	74.3 ± 8.7	74.1 ± 7.5	73.5 ± 9.7	74.1 ± 8.4	75.6 ± 9.0	0.525
WBC (×10^3^/μL)	4760 ± 1132	4700 ± 1081	4701 ± 1094	4770 ± 1278	4871 ± 1084	0.795
RBC (×10^6^/μL)	496 ± 35	501 ± 33	490 ± 33	496 ± 38	497 ± 34	0.362
Hb (g/dL)	15.0 ± 0.8	15.1 ± 0.8	15.0 ± 0.9	15.0 ± 0.9	15.1 ± 0.7	0.644
Ht (%)	46.6 ± 2.5	46.6 ± 2.4	46.3 ± 2.7	46.7 ± 2.7	46.7 ± 2.4	0.789
PLT (/μL)	22.6 ± 4.6	22.3 ± 4.6	23.4 ± 4.6	21.7 ± 4.8	22.8 ± 4.3	0.143
T-CHO (mg/dL)	196 ± 30	200 ± 34	198 ± 28	192 ± 29	193 ± 28	0.355
HDL-C (mg/dL)	60 ± 14	58 ± 14	60 ± 14	61 ± 13	60 ± 15	0.565
LDL-C (mg/dL)	114 ± 28	118 ± 28	117 ± 26	111 ± 30	112 ± 27	0.369
TG (mg/dL)	75 (56, 100)	89 (57, 119)	79 (57, 95)	73 (56, 93)	66 (51, 89)	0.050
AST (IU/L)	24 ± 16	27 ± 30	22 ± 5	23 ± 8	22 ± 6	0.187
ALT (IU/L)	22 ± 11	25 ± 13	20 ± 9	22 ± 13	20 ± 10	**0.041**
γ-GTP (IU/L)	36 ± 27	42 ± 34	30 ± 16 *	39 ± 28	32 ± 27	**0.027**
BUN (mg/dL)	14 ± 3	13 ± 3	14 ± 3	15 ± 3	15 ± 4 *	**0.007**
Cr (mg/dL)	0.97 ± 0.12	0.97 ± 0.10	0.96 ± 0.12	0.95 ± 0.14	1.01 ± 0.12	0.056
UA (mg/dL)	6.5 ± 1.3	6.6 ± 1.3	6.3 ± 1.2	6.4 ± 1.3	6.7 ± 1.3	0.297
FBS (mg/dL)	97 ± 9	98 ± 8	97 ± 8	98 ± 8	97 ± 11	0.752
IRI (μU/mL)	4.9 ± 2.0	5.2 ± 2.2	4.7 ± 1.8	5.1 ± 2.2	4.9 ± 2.0	0.459
HOMA-IR	1.2 ± 0.5	1.3 ± 0.6	1.1 ± 0.5	1.2 ± 0.6	1.2 ± 0.5	0.432
HOMA-beta	53 ± 21	54 ± 20	50 ± 17	53 ± 22	54 ± 24	0.652
HbA1C (%)	5.7 ± 0.3	5.7 ± 0.3	5.7 ± 0.3	5.7 ± 0.3	5.7 ± 0.4	0.980
CRP (mg/dL)	0.09 (0.08, 0.12)	0.10 (0.09, 0.14)	0.09 (0.09, 0.11)	0.10 (0.08, 0.14)	0.09 (0.08, 0.13)	0.076
**Previous History (Under treatment)**						
Hypertension, *n* (%)	11 (4)	3 (4)	3 (4)	4 (6)	1 (2)	0.606
Hyperlipidemia, *n* (%)	10 (4)	4 (6)	4 (6)	2 (3)	0	0.216
Hyperuricemia, *n* (%)	24 (9)	9 (13)	5 (7)	7 (10)	3 (4)	0.306
Metabolic Syndrome, *n* (%)	12 (4)	3 (4)	6 (9)	0	3 (4)	0.106
NAFLD, *n* (%)	63 (23)	14 (20)	15 (22)	10 (27)	16 (25)	0.820
Alcohol intake (g/day)	1.14 (0.57, 2.29)	1.14 (0.58, 2.46)	1.20 (0.57, 2.49)	1.14 (0.57, 2.00)	1.03 (0.41, 2.57)	0.430
**Smoking Status**						
Non-smoker, *n* (%)	94 (34)	21 (30)	27 (40)	27 (40)	19 (28)	**0.003**
Current smoker, *n* (%)	71 (26)	10 (15)	15 (22)	17 (25)	29 (43)	
Past smoker, *n* (%)	108 (40)	38 (55)	27 (39)	23 (34)	20 (29)	

The subjects were divided into the quartile groups by urine indole concentration level. BMI, body mass index; WBC, white blood cells; RBC, red blood cells; Hb, hemoglobin; PLT, platelets; LDL, low density lipoprotein; TG, triglyceride; HDL, high density lipoprotein; AST, aspartate aminotrans-ferase; ALT, alanine aminotransferase; γ-GTP, γ-glutamyltransferase; BUN, blood urea nitrogen; Cr, creatinine kinase; UA, uric acid; FBS, fasting blood sugar; IRI, immunoreactive insulin; HOMA, homeostasis model assessment; IR, insulin resistance; HbA1C, hemoglobin A1C; CRP, C-reactive protein; NAFLD, non-alcoholic fatty liver disease. * *p* < 0.05 vs. Q1. Bold values show *p* < 0.05.

**Table 2 metabolites-12-00234-t002:** Univariate and multivariate regression analysis for urine indole levels.

*n* = 273	Univariate		Multivariate			
			Model 1		Model 2	
	β	*p*	β	*p*	β	*p*
Age (years)	−0.061	0.315	−0.063	0.307		
BMI (kg/m^2^)	0.002	0.974	−0.007	0.918		
TG (mg/dL)	−0.130	**0.031**	−0.095	0.121	−0.116	0.051
ALT (IU/L)	−0.120	**0.047**	−0.090	0.160		
BUN (mg/dL)	0.148	**0.014**	0.116	0.060	0.152	**0.011**
Cr (mg/dL)	0.121	**0.045**	0.096	0.120		
Hypertension, yes 1, no 0	0.148	0.015	0.024	0.690		
Hyperlipidemia, yes 1, no 0	0.109	0.072	0.092	0.122		
NAFLD, yes 1, no 0	−0.036	0.548	−0.087	0.203		
Alcohol intake (g/day)	−0.087	0.158	−0.085	0.156		
Current-Smoking, yes 1, no 0	0.160	**0.008**	0.155	**0.011**	0.173	**0.004**

The adjusted R2 for model 1 and model 2 were 0.059 (p = 0.003) and 0.055 (*p* < 0.001), respectively. Abbreviations are the same as in Table 1. β: standardized coefficient. Bold values show *p* < 0.05.

**Table 3 metabolites-12-00234-t003:** Diurnal variations and effect of smoking on urine indole levels.

Participants	Time 1	Time 2	Time 3
A1	22.52	22.97	23.21
A2	39.01	37.89	38.34
A3	12.55	40.22	28.00
A4	29.62	35.48	44.20
B1	28.57	33.94	33.69
B2	46.70	34.13	30.02
B3	44.36	58.36	60.86
B4	37.69	28.65	14.91
C1	51.44	51.43	65.79
C2	52.35	46.56	29.54
C3	42.67	42.68	45.85
C4	35.23	36.35	28.33

The unit of indole concentration is mg/L. A1–4: paper cigarette smokers, B1–4: electric cigarette smokers, C1–4: non-smokers. Time 1: night before sleep, Time 2: morning before smoking, Time 3: several hours later after Time 2 (for smokers, after smoking). The interval from Time 1 to Time 2 and from Time 2 to Time 3 was 7.0 ± 1.5 h and 2.1 ± 0.5 h, respectively. The interval from the first smoking on Day 2 (after Time 2) to Time 3 was 1.5 ± 0.8 h.

## Data Availability

Date is contained within the article or Appendix A.

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
