# Peer review of "Association between Smoking and Urine Indole Levels Measured by a Commercialized Test"

_metabolites, 2022, doi:10.3390/metabo12030234_

Round 1

Reviewer 1 Report

In this manuscript, Mine et al presented their study on urinary indole levels measured by a commercialized kit using Kovacs regent and smoking status in 273 healthy men. Their results show that smoking is associated with increased levels of urine indole. My comments are given below.

Major: based on the study context, there are three main factors in this study: indole, smoking status and gut microbiome. We want to study their relationships, we should try our best to measure them as best as we can. However, only indole is measured objectively by commercialized kit. The smoking status is self reported. It seems natural to also quantitatively and objectively measure smoking status, such as urine nicotine / cotinine level. This will give more weight to the conclusion and is quite relevant when comparing paper- and e-cigaret.  While gut microbiome is not measured at all but discussed extensively throughout the manuscript. It will be very informative to study gut microbiome profiles and links the data with the indole level. Overall, I would encourage the authors to provide more data to enable more in-depth inquiry of this very topic. 

Minor: 

1) P12: tryptophanase-positive bacteria

2) P174: cress-section => cross-sectional

3) Please provide more details or parameters for univariate and multivariate regression analyses

Author Response

Thank you very much for your suggestions. We have read the comments carefully and improved the manuscript according to your suggestions. We hope you will revise it to the new version. Thank you very much for your review.

Comments and Suggestions for Authors

In this manuscript, Mine et al presented their study on urinary indole levels measured by a commercialized kit using Kovacs regent and smoking status in 273 healthy men. Their results show that smoking is associated with increased levels of urine indole. My comments are given below.

Major:

  1. based on the study context, there are three main factors in this study: indole, smoking status and gut microbiome. We want to study their relationships, we should try our best to measure them as best as we can. However, only indole is measured objectively by commercialized kit. The smoking status is self reported. It seems natural to also quantitatively and objectively measure smoking status, such as urine nicotine / cotinine level.
  • Response; Thank you very much. Smoking cigarettes produces a lot of indole derivatives. There are many papers that deal with them. However, their measurement requires a lot of money, time, and effort, and cannot be used in a clinical setting. Commercially available kits can only detect pure indole (apart from specificity), but I think the convenience is worth it. Since the results were obtained in an exploratory manner, it was not possible to measure indole in urine along with nicotine. The relationship between indole and tobacco was not anticipated. Therefore, we have added it as a limitation.
  • Improvement;
    • Introduction, Page2 Line53-55; We changed a sentence to “Although a liquid/mass spectrometer can provide precise information of metabolites in urine, the method needs time and effort, and cannot be used in a clinical setting. The convenient test might be useful as a screening tool.”
    • Limitation, Page9 Line229-230; We added “It is ideal to measure smoking status objectively, such as urine nicotine / cotinine level. However, the assessment could not be performed in the study.”

  1. This will give more weight to the conclusion and is quite relevant when comparing paper- and e-cigarette. While gut microbiome is not measured at all but discussed extensively throughout the manuscript.
  • Response; I agree with your comment. We have changed the manuscript to equally address changes in gut microbiota and the potential for absorption of tobacco products. We would appreciate it if you could revise the new Introduction and Discussion.
  • Improvement;
    • Abstract, Introduction, Page1 Line 13-14; We added the sentence “Cigarette smoke contains indoles which could also change the urine indole levels.”
    • Introduction, Page2 Line 37-38; We introduced that the tobacco contains indole.
    • Discussion, Page8 Line 193-211; We created a new section about the possibility of cigarette smoke for the cause of results.

  1. It will be very informative to study gut microbiome profiles and links the data with the indole level. Overall, I would encourage the authors to provide more data to enable more in-depth inquiry of this very topic.
  • Response; Thank you very much. We would like to try it. However, it may take another couple of years. This time we would like to open the acquired results.
  • Improvement; We are very sorry for that we could not add experiments. We claims “A direct comparison of urine indole levels with fecal amount of indole and tryptophanase-positive bacteria is necessary to prove the involvement of intestinal microflora in future.” in Page9 Line 215-217.

Minor:

1) P12: tryptophanase-positive bacteria

2) P174: cress-section => cross-sectional

  • Response; Thank you very much. We corrected the mistakes.
  • Improvement; Page 8 Line178; cross-sectional”, Page 1 line 12; “tryptophanase-positive bacteria”

3) Please provide more details or parameters for univariate and multivariate regression analyses

  • Response; Thank you very much. Data on urinary indole concentration and other clinical factors are disclosed in the Supplementary file. However, there are no factors associated with urinary indole concentration.
  • Improvement; We made Table S2 in the Supplemental File. It was indicated in the main text (Page6 Line122).

Reviewer 2 Report

The manuscript under review “Association of smoking with urine indole levels measured by a commercialized test” for the journal of metabolites is interesting but has several flaws. I mentioned these flaws below:

  1. In the paper the authors show changes in indole in the urine and they claimed that the changes in the urine indole are due to the changes in microbiota but there is very sparse evidence was provided.
  2. The Second paragraph of the introduction is not needed. It is a distraction. This paragraph talks about TAMO, which is a gut-associated metabolite but is not an indole and there is no use of this paragraph in describing the story they have in hand.
  3. Please keep all the bacterial names in italics (Line 48, 62, 165, 168, 173,188)
  4. Line 74: Please provide the full name of BUN when you are describing it for the first time.
  5. In Figure 1: Based on the data and explanation, we would assume a large section of non-smokers should be in Q1 but the larger portion of non-smokers belonged to Q2 and Q3. I did not find any explanation for such discrepancy in the manuscript. Why a P-value is written on the top of the figure; is nowhere explained.
  6. Figure 2: There is a P-value written on the top of the figure but there is nowhere explained in the text. Please explain what is on the figure.
  7. Please explain the toxins in cigarettes and especially when they have indole-containing toxins (PMID: 15716489).
  8. The paper claims that smokers' urine has more indole that they can measure and possibly tell who smokes or not if smokers have more indole due to changes in microbiota, which could happen by several means, how one could determine that the increase of indole in the urine is due to cigarettes and not due to changes in microbiota by other means. Few indoles are generated in cigarette smoke, and authors cannot rule out these indoles in the urine.

Author Response

Thank you very much for your suggestions. We have read the comments carefully and improved the manuscript according to your suggestions. We hope you will revise it to the new version.

Comments and Suggestions for Authors

The manuscript under review “Association of smoking with urine indole levels measured by a commercialized test” for the journal of metabolites is interesting but has several flaws. I mentioned these flaws below:

  1. In the paper the authors show changes in indole in the urine and they claimed that the changes in the urine indole are due to the changes in microbiota but there is very sparse evidence was provided.

  • Response; We agree with your comment. This problem was also pointed by the reviewer 1. The study was started in an exploratory manner. The relationship with smoking was discovered expectedly. We have changed the manuscript to equally address changes in gut microbiota and the potential for absorption of tobacco products. We would appreciate it if you could revise the new Introduction and Discussion.
  • Improvement;
    • Abstract, Introduction, Page1 Line 13-14; We added the sentence “Cigarette smoke contains indoles which could also change the urine indole levels.”
    • Introduction, Page2 Line 37-38; We introduced that the tobacco contains indole.
    • Discussion, Page8 Line 193-211; We created a new section about the possibility of cigarette smoke for the cause of results.

  1. The Second paragraph of the introduction is not needed. It is a distraction. This paragraph talks about TAMO, which is a gut-associated metabolite but is not an indole and there is no use of this paragraph in describing the story they have in hand.
  • Response; I delated the section in the introduction.

  1. Please keep all the bacterial names in italics (Line 48, 62, 165, 168, 173,188)
  • Response; Thank you very much. We changed them to be in italics.
  • Improvement; We used italic for bacteria name (Line 33, 172,183-184,188-189, 254)

  1. Line 74: Please provide the full name of BUN when you are describing it for the first time.
  • Response; Thank you very much. We spelled it out at the first appearance.
  • Improvement; Page2 Line64 ” blood urea nitrogen (BUN)”

  1. In Figure 1: Based on the data and explanation, we would assume a large section of non-smokers should be in Q1 but the larger portion of non-smokers belonged to Q2 and Q3.
  • Response: Thank you very much. I agree with your comment. It is true that there were non-smokers who had high urinary indole levels. The reason for this was not elucidated this time, and the existence of other factors is expected.
  • Improvement;
    • Discussion, Page7 Line151-154; We added the sentences “Some participants without smoking habits had high urinary indole concentrations whereas the ratio of past-smokers decreased with the rank of quartiles. The reason was unknown but existence of influencing factors other than smoking was suggested.”

  1. I did not find any explanation for such discrepancy in the manuscript. Why a P-value is written on the top of the figure; is nowhere explained. Figure 2: There is a P-value written on the top of the figure but there is nowhere explained in the text. Please explain what is on the figure.
  • Response: We added the explanation of the P-value in the footnotes.
  • Improvement;
    • Figure1 footnote, Page4 Line84-85;” The P-value on the graph stands for the statistical significance of difference in the proportion of the three smoking groups among the quartile groups (P=0.003).”
    • Figure2 footnote, Page5 Line102-103; “The P-value on the graph stands for the statistical significance of difference among three groups (P=0.001).”
    • Figure3 footnote, Page6 Line113-114;” The P-value on the graph indicates that there is not significant difference among the groups.”

  1. Please explain the toxins in cigarettes and especially when they have indole-containing toxins (PMID: 15716489).
  • Response: We really appreciate it that you introduced us the very important paper. We changed the discussion and created the new section about the absorption of indole from cigarette smoke. The improvement was done based on the paper.
  • Improvement;
    • Discussion, Page8 Line 194; The study was referred in the paragraph.

  1. The paper claims that smokers' urine has more indole that they can measure and possibly tell who smokes or not if smokers have more indole due to changes in microbiota, which could happen by several means, how one could determine that the increase of indole in the urine is due to cigarettes and not due to changes in microbiota by other means. Few indoles are generated in cigarette smoke, and authors cannot rule out these indoles in the urine.
  • Response: Thank you very much. The answer for the comment was same as that for the comment 1.

Reviewer 3 Report

Presented manuscript No. “metabolites-1581775” describes quantification of indole level in urine of smokers to asses changes intestinal microflora among smokers. Authors used commercially test kit with Kovac reagent. The work is very interesting. Generally, most of the paper is well constructed (typically for scientific manuscripts), starting with introduction, materials and methods, results and discussion. The Authors widely and properly described current state of knowledge. However, some parts of the paper were described not clearly (without important details). There are also some editing errors. Therefore, some issues should be clarify and explain (some more serious and some minor) to make this paper proper for publish.

  • There are many spacebars missed. Please verify whole text in this context.
  • Abstract L. 15: missed spacebar. Were there any criteria for selecting the research group (also according to Table 1)? why people of this age were selected?
  • 30-45 – This fragment is not directly associated with main aim of the paper and research. In my point of view this fragment should be shortened, rephrased or deleted. Instead of this fragment, more information about indole and associated disorders should be added.
  • 57: “mass spectogram” is not very professional/scientific term. It should be “mass spectrum”
  • 60-68: Authors stated that LC-MS-based methods are preferred for quantification of indole level in urine, but colorimetric tests are also proper as screening methods. However, in studies Authors used such colorimetric test for quantification. The information provided does not match. This fragment is not clear. Please rephrased and explain.
  • Analytical procedure should be described. Moreover, chemicals, reagents and their purity should also be listed like for typical scientific manuscripts.

Based on the Fig. 2, I am not sure that level of indole is different among non-smokers and smokers (Box plots/ whiskers are covering). Of course, median values are various, but this issue requires explanation.

Author Response

Thank you very much for your suggestions. We have read the comments carefully and improved the manuscript according to your suggestions. We hope you will revise it to the new version. 

Comments and Suggestions for Authors

Presented manuscript No. “metabolites-1581775” describes quantification of indole level in urine of smokers to asses changes intestinal microflora among smokers. Authors used commercially test kit with Kovac reagent. The work is very interesting. Generally, most of the paper is well constructed (typically for scientific manuscripts), starting with introduction, materials and methods, results and discussion. The Authors widely and properly described current state of knowledge. However, some parts of the paper were described not clearly (without important details). There are also some editing errors. Therefore, some issues should be clarify and explain (some more serious and some minor) to make this paper proper for publish.

  1. There are many spacebars missed. Please verify whole text in this context. Abstract L. 15: missed spacebar.
  • Response; Thank you very much. We corrected the mistakes.
  • Improvement; The former sentence at Line15 was already deleted for another reason.

  1. Were there any criteria for selecting the research group (also according to Table 1)? why people of this age were selected?
  • Response: This survey was conducted by the Air Self-Defense Force during their annual health checkup. The number of people required to undergo the physical examination was determined by their age. However, we believe this is an advantage of this study because it eliminates other clinical factors.
  • Improvement;
    • Method, Page9 Line248; we added “40-55 years old”.
    • Limitation, Page9 Line238-239; We add “However, the restricted study population is strength for elucidating effect of smoking by excluding other clinical factors.”

  1. 30-45 – This fragment is not directly associated with main aim of the paper and research. In my point of view this fragment should be shortened, rephrased or deleted. Instead of this fragment, more information about indole and associated disorders should be added.
  • Response; Thank you very much. I delated the section in the introduction.
    • Introduction; The paragraph of TMAO, another example of biomarker, was deleted.

  1. 57: “mass spectogram” is not very professional/scientific term. It should be “mass spectrum”
  • Response; Thank you very much. I corrected the words.
  • Improvement; We used “spectrum” (Line 43, 241)

  1. 60-68: Authors stated that LC-MS-based methods are preferred for quantification of indole level in urine, but colorimetric tests are also proper as screening methods. However, in studies Authors used such colorimetric test for quantification. The information provided does not match. This fragment is not clear. Please rephrased and explain.
  • Response; Thank you very much. I corrected the sentences.
  • Improvements; We changed the sentence to “Although a liquid/mass spectrometer can provide precise information of metabolites in urine, the method needs time and effort. The convenient test might be useful as a screening tool.” Line53-55

  1. Analytical procedure should be described. Moreover, chemicals, reagents and their purity should also be listed like for typical scientific manuscripts.
  • Response; Since this is a commercialized product, the manufacturer keeps the information confidential. We asked it to the manufacture but could not have the permission yet. However, we have explained the procedure and more as best we can.
  • Improvement; We added explanation for the measurement procedure. Page10 Line269-278.

  1. Based on the Fig. 2, I am not sure that level of indole is different among non-smokers and smokers (Box plots/ whiskers are covering). Of course, median values are various, but this issue requires explanation.
  • Response; We checked the statistics and found that the results were the same. Therefore, the actual p-values are given in the text.
  • Improvement;
    • Results, Page4 Line92-93; We added the p-values for the difference in each combination of the groups.

Round 2

Reviewer 1 Report

The authors have made sufficient revisions to improve the manuscript. I have no more comments. 

Reviewer 2 Report

Authors have answered all the questions best of their abilities. 

Reviewer 3 Report

I accept review.